# What Is the Shape of Geographical Time-Space?
# A Three-Dimensional Model Made of Curves and Cones

**Alain L'Hostis** [1,*] and **Farouk Abdou** [2]

1   LVMT, Université Gustave Eiffel, IFSTTAR, Ecole des Ponts, F-77454 Marne-la-Vallée, France
2   Ministère de la Défense, 75007 Paris, France; bobactor40@gmail.com
*   Correspondence: alain.lhostis@univ-eiffel.fr

**Abstract:** Geographical time-spaces exhibit a series of properties, including space inversion, that turns any representation effort into a complex task. In order to improve the legibility of the representation and leveraging the advances of three-dimensional computer graphics, the aim of the study is to propose a new method extending time-space relief cartography introduced by Mathis and L'Hostis. The novelty of the model resides in the use of cones to describing the terrestrial surface instead of graph faces, and in the use of curves instead of broken segments for edges. We implement the model on the Chinese space. The Chinese geographical time-space of reference year 2006 is produced by the combination and the confrontation of the fast air transport system and of the 7.5-times slower road transport system. Slower, short range flights are represented as curved lines above the earth surface with longer length than the geodesic, in order to account for a slower speed. The very steep slope of cones expresses the relative difficulty of crossing terrestrial time-space, as well as the comparably extreme efficiency of long-range flights for moving between cities. Finally, the whole image proposes a coherent representation of the geographical time-space where fast city-to-city transport is combined with slow terrestrial systems that allow one to reach any location.

**Keywords:** geographical time-space; transport; cartography

## 1. Introduction

Geographic space is known through experience and its narration, and through representation. Geographic distances are experienced through the use of means of transport, while representations usually take the form of maps. In the domain of the representation of geographic space, even considering the earliest maps [1], many efforts have focused on improving the coherence between the experience of space through movement and the representation itself. Most of these proposals introduced deformations of the conventional map [2–6] and belong to the family of anamorphic cartography. Other proposals have focused on network representation [7–9] with springs more or less compressed to express the temporal length of routes. These representations have issues that we will expose below in Section 4.

Time geography [10] and its developments in GIScience [11–13] indicate another stream of investigation of geographical time-space but more focused on understanding and modelling movement and its constraints than on producing a new representation of geographical time-space. In particular, time geography contains no significant development regarding geographical distances. In a broader scope, cartographers have started importing methods developed in the field of scientific visualisation [14,15]. This stream of "visual analytics" [16,17] aims at implementing improved graphical outputs to make complex spatial data intelligible. While our datasets are comparably much more simple than what is generally used in visual analytics, we also make use of sophisticated visual representation methods borrowed from scientific visualisation.

More generally, beyond these cartographic and visualisation works, in geography and related fields, only a few efforts have been dedicated to the study of distances [18–24].

This observation is also valid in regards to time geography and visual analytics that both dedicate most efforts on movement data rather than on distances.

We propose in this paper a geographical time-space model extending time-space relief cartography introduced by Mathis and L'Hostis [25–28]. We will expose the principles of the model and, in the annexes, detail the mathematics needed for graphical implementation.

Furthermore, we want to produce a cartographic model of current geographical time-space. In this aim we start with the study of the properties of geographical time-space transport modes. After an empirical section, we draw a state of the art of cartographic representation of geographical time-space. Finally, we introduce our model.

## 2. The Properties of Geographical Time-Space

We call geographical time-space a representation of the geographical space expressing the time spent to reach places. Representations of geographical time-space usually come with a timescale. The study of geographical time-space reveals a series of properties.

Firstly, geographers observed, at least since antiquity [29] (p. 19-3), an acceleration of human movement, and expressed the idea of the shrinking of geographical time-space [3,30].

Secondly, the crossing of geographical space is performed by a range of coexisting transport means. In the current transport system, several transport means with distinct characteristics and especially distinct speed levels, can be observed. The principle of coexistence of several transport modes can also be traced back in the history of transport [31]. Each transport system has its own domain of relevance, and the principle of their co-existence is a key feature of geographical time-space. However this represents a critical challenge for the representation of time-space and geographical distances due to the coexistence of different speeds.

Thirdly, movements in the geographical space are mostly realised by means of transport networks. This means that the description of geographical time-space should exhibit network properties. Transport networks may be continuously or discontinuously accessible. The usual road network or the urban public space are an example of continuously accessible networks, as opposed to the cases of expressways, railways, or airlines that are only accessible through access points. The other key properties of networks relates to the dialectic of straight lines and detour [32]. Paths through networks follow predefined lines and almost always escape the straight line between origin and destination.

Finally, the most geometrically disturbing property of geographical time-spaces derives from an exacerbation of network configurations exposed before. Space inversion [33,34] occurs when the initial portion of a trip goes in the opposite direction to that of the end destination. It is abundantly observed in geographical time-space and occurs in the proximity of access-point of discontinuous networks such as expressway entrances, railway stations, and airports. In these cases the order of proximities in geographical time-space is profoundly disturbed. As shown in Figure 1, a place B located in-between two other places, A and C, in geographical space may have a different position in geographical time-space: C is closer to A (1 h) than to B (1 h 20 min) in time-space.

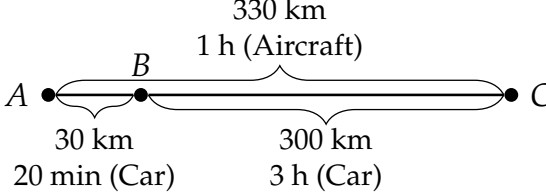

**Figure 1.** Space inversion between three places located in geographical space (kilometres) and in geographical time-space (duration).

It can be observed that these four properties of geographical time-spaces are tightly linked to each other. Thus, space inversion is generated by the design of transport networks,

and implies the principle of coexistence of several transport means. After this first section on the properties of geographical time-space, we will expose our empirical approach of the measurement of transport speed.

### 3. The Speed of Air and Road Transport Systems

The air transport system lies at the top of the current hierarchy in terms of speed level. The top speed of a typical jet aircraft is about 900 km/h which converts into a typical commercial speed of 750 km/h if we consider for example a current London to New York flight, which is 7 h 45 min.

We collected data on a sample of air routes with great circle kilometre distance and duration from timetable (we extracted data from the website flightaware.com in January 2016). Long distance east-west flights exhibit significantly different schedules according to the direction of flight regarding trade wind. However the model needs symmetrical data for the network. In this case we chose as the duration, the greatest measure, generally from west to east.

The analysis of current timetables on a sample origin-destination pairs in Figure 2 shows that, in terms of speed, we can consider two situations. Below 2000 km in length, the speed of services can be approximated with a linear function, $s = (s_{long-range}/2000) * l = 0.375 * l$, where $l$ is the length of the service between origin and destination following the geodesic line. Beyond 2000 km of service length, we approximate speed with the value of 750 km per hour. Further developments could introduce an exponential function in order to better represent the empirical data.

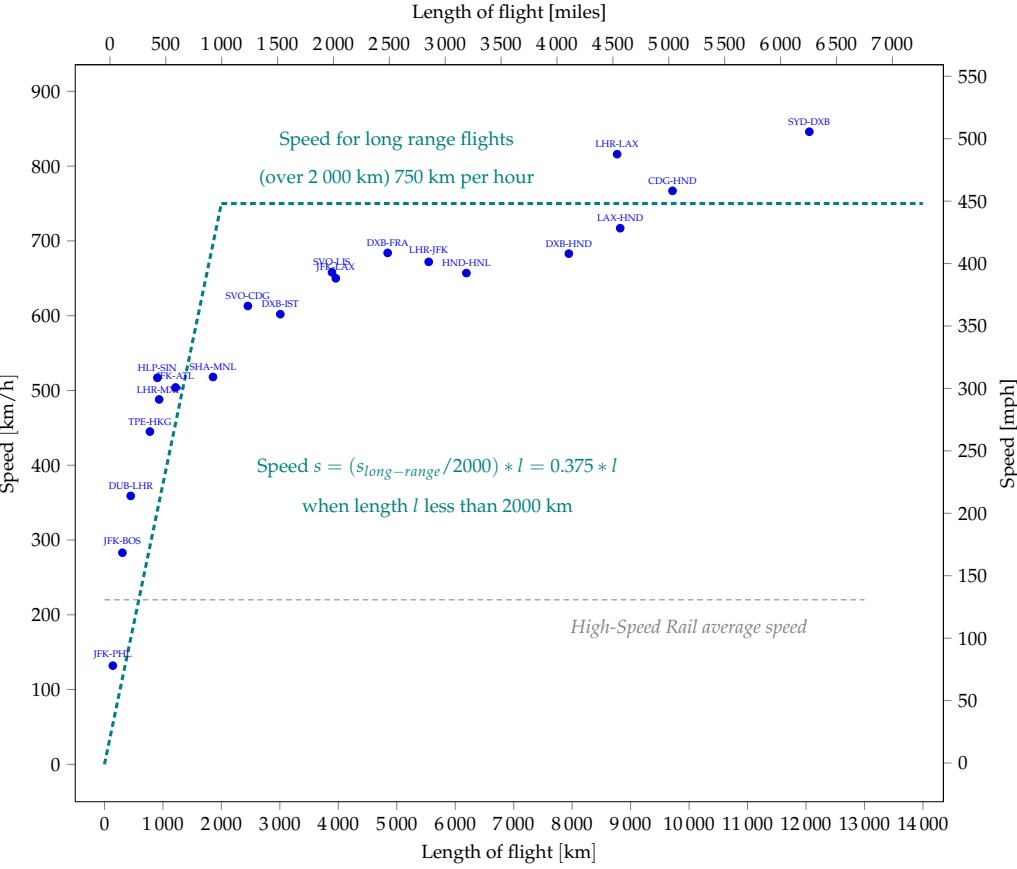

**Figure 2.** The commercial speed of aircraft services on a sample of origin destination pairs (data from www.flightglobal.com in 2016) and a linear approximation.

According to the principle of coexistence of transport modes, in complement to the air transport system, reaching destinations necessitates the use of terrestrial transport

systems. The terrestrial transport system can be described with a unique speed of 100 km per hour [35]. This value represents an approximation of car speed over long distance on motorways, as well as train speed on classical rail systems.

Once the key parameters are fixed we can start considering the issue of the representation of geographical time-space.

## 4. Cartographic Representations of Geographical Time-Space

The improvement of transport means that over time it has had profound influence on our representation of geographical space. The more common sense idea, the first property of geographical time-spaces, states that the world is getting smaller. Nevertheless, the idea of a uniform shrinkage [36,37] has been challenged by geographers [5,7,30]. The main critic issued against the principle of uniform shrinkage is the idea that the entire intermediate and surrounding geographical space benefits from transport conditions improvements in a given area or between two locations. These representations do not consider the widely accepted fact that geographical time-space is not homogeneous [5].

More generally, the issue of the cartographic representation of time-space can be seen as transforming a kilometre-space into a time-space. In the cartographic representation, we will then replace the usual and conventional kilometre scale by a timescale.

The various methods for the production of cartographic time-space can be subdivided into two categories [38,39] (p. 90). Most methods found in the literature move the position of places on the map in order to reflect time-space distances. Many representations of this type were introduced during the plastic spaces period in the history of geography [39] (p. 90), in reference to the expression proposed by Forer [4] to describe a geography freed from cartographic conventions. Since this very active period, regularly new implementations emerge [40–43]. Cartographic representations attached to this movement are of the type of the anamorphic cartography [33,44]. In plastic space maps, the heterogeneous shrinkage of space is assumed. However these maps imply that all the space located in-between places that are moved closer to each other benefit from the time-space contraction. The fact that these maps do not account for the phenomenon of space inversion (see Figure 1), a key feature of geographical time-spaces, is the main critic formulated against this type of representation [28].

The second category of geographical time-space representation consists in preserving the location of places, and in drawing the edges in a way that expresses time-distances. Janelle introduced curved lines in network style diagrams to account for lower speeds of slow transport systems [7]. This approach produced a coherent representation of time-distances with multiple speed. Tobler proposed sketch maps where compressed springs express variations in the efficiency of edges in a network [9]. These two first proposals are relevant as network representations, but fail at representing space in a continuous way. Overcoming this limitation, and belonging also to this category, time-space relief cartography were conceptualised by Mathis [25] and implemented by L'Hostis [26].

The time-space relief cartography is based on three general principles:

- Cities, considered as nodes of the transport network, remain at their conventional geographical location.
- Edges are drawn in the third dimension, proportionally to the travel time needed between nodes.
- The geographic surface is attached to the slowest network, i.e., the road network.

These principles imply that the height of cities does not have the meaning it has in classical thematic cartography. Proportional time-distances is the only quantity that the map conveys, by design.

When drawing edges in the three-dimension time-space representation, one principle must be observed, but several rules can be chosen. The principle of the representation implies that edge length must be proportional to the associated time-distance.

The shortest edges must be drawn as straight lines in the plane, or as geodesic curves on the sphere. All the other edges, characterised by a slower speed than the fastest existing

speed in the considered space, must be drawn longer than the fastest edges, proportionally to the longer time needed to join the two location. This type of cartography permits, by design, the simultaneous representation of fast and slow transport modes. The design of the model conveys a significant level of abstraction regarding the actual path of connections or links between cities for all transport modes. For instance, while air connection rarely follows a straight line or the geodesic [45], our model takes for reference the most direct route in order to allow for visually reading lengths that converts in time.

Providing the principle of proportionality is respected, a degree of freedom is allowed in drawing the edges between two nodes of the graph. The simplest way consists in drawing two segments with a middle point located under the surface of the earth. If the time-space structure is an obstacle to the drawing of slow edges, it is possible to draw the edges above the surface of earth [46]. From the representation of terrestrial transport modes edges, it is possible to infer a transport surface.

Initially applied only to terrestrial transport modes, the model was extended in the 2000s to the air transport system, on continental spaces [28]. The next step is the current proposal that develops a new design for the transport surface, and introduces curves for long edges. While in previous time-space cartographic designs the terrestrial transport surface was based on the faces of a graph, we introduce the use of cones.

## 5. The Principles of A Representation of Geographic Time-Space Based on Three-Dimensional Cones

Unlike all previous time-space relief representation [26–28,47] where the geographical surface is based on the edges of a graph, we introduce conic shapes to represent terrestrial time-space. Each city and its surrounding space are then represented by a cone. The spiky edge of the cone ends at the usual geographic location of the city, and the cone geometry extends below the earth surface. The terrestrial time-space is formed by the three-dimensional surface resulting from the aggregation of all the cones together.

The use of cones allows for generalising the terrestrial transport surface and simplifying the principle of construction of the three-dimensional model. In addition, the design based on cones permits to avoid the random slopes generated by the three-dimensional facets and replaces them by a uniform, controlled slope. A uniform slope for the terrestrial transport surface is, from a theoretical point of view, more satisfactory than previous three-dimensional network based facets [28] where the slope of facets was not fully controlled, as it is more coherent with the basic principle of the representation which indicates a unique slope for a given transport mean.

The cone is the basic structure of this representation. Cones are characterised by a slope that follows the ratio between the speed of basic terrestrial networks, i.e., the road system, and the maximum available speed, which is attained, on the period that we consider, through the air transport system.

In Figure 3, two cities, *a* and *b*, are joined by a fast non-stop transport system (in red), and also joined by a slower terrestrial transport mode (in blue). The fast transport link is represented through a straight line of speed $s_{max}$, while the cones represent the surface supporting the slower terrestrial transport mode of speed $s_{amb}$, with *m* as the midpoint between the two cities and the intersecting point of the two cones. Assuming that terrestrial travel speed are similar in all the geographic space considered, the slope of the cones centred on the two cities is a coherent representation of time-space. In Figure 3, the slope of the cones implies that the length of the straight red line, and the length of the blue line drawn on the cone and joining the two cities, are proportional to the respective travel times, according to a provided formula of a ratio of speed. This three-dimensional geometry forms a time-space representation with different transport speed.

The mathematical formulae of the representation are presented in the annexes of the paper: The geometry of the cones (Appendix A), the geometry of edges in spherical geometry (Appendix B), and the geometry of smoothed edges in projected geometry (Appendix C).

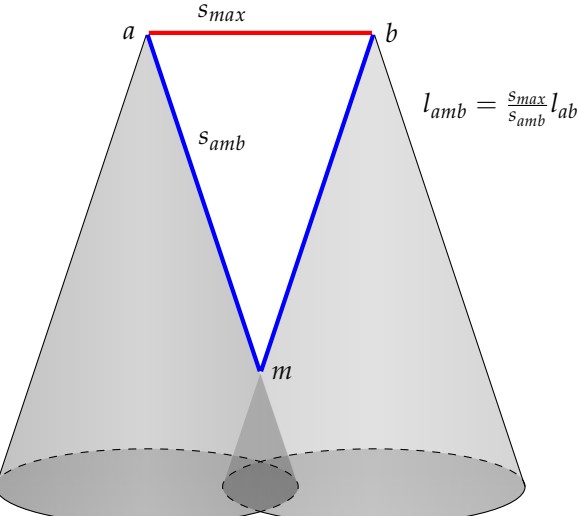

**Figure 3.** Cones and edges in three dimensions as a basic structure for time-space representation.

## 6. Theoretical Validation

We can test the ability of the representation to respect and render the four properties introduced in Section 2.

The property of acceleration may be rendered by producing side by side two states of the representation at two different periods where transport speed differ. A common timescale will help to express the transformations provoked by an acceleration of transport speed. Hence, the representation is able to account for the acceleration of human movement over historical time.

Secondly, the coexistence of transport means is introduced by design since the representation is based on a graph of the transport network characterised by several transport modes.

Thirdly, the combination of continuous and discontinuous access is provided by the design of the three-dimensional structure: The cones represent the continuous geographic surface, the lines represent the links of the transport network, and the summit of cones, where the geographical surface and fast transport network meet, represent the cities. In the representation, the two modes of representation of geographical time-space–namely continuous and network based discontinuous–coexist and are attached to each other through cities.

Finally, the space inversion is made intelligible and readable on the representation: From a point located on the slope of a cone, close to the top, the fastest path to a distant city implies going back up to the nearby city, and taking the fast transport mode.

The following steps of the research will involve user tests in order to provide practical validation. They extend beyond the scope of the current article.

## 7. Dataset

In order to build a map of the geographical time-space, we firstly need to establish a list of cities. We used the 2014 revision of the World Urbanisation Prospect by the United Nations [48] as it was the most recent high quality urban dataset available at the time of creation of the representation.

The air services network is derived from the data provided by openflights.org (https://openflights.org/data.html (accessed on 13 September 2018)). The network of flights is from 2006, the most recent dataset available from this source. In addition, the aerial transport system of 2006 can be considered as representative of the 2014 condition. Links in the network correspond to air routes served daily by at least three aircraft of a capacity of more than 70 passengers. We defined this threshold to consider only origin-destinations pairs with a significant service available.

Tests, which can be seen at the blog https://timespace.hypotheses.org (accessed on 13 September 2018), shown that the shorter the list of cities, the easier it is to produce a

readable representation. In this view, we used two criteria to operate the selection of cities. Only cities with more than a million inhabitants were considered. In addition, we chose only those cities served by aerial services in 2006. This selection rules applied to the UN cities database generated a list of 45 Chinese cities, including Taiwan.

## 8. Implementation of the Geographical Time-Space of China

The three-dimensional model on Figure 4 was produced by the *Shriveling world* software developed by the authors and B. Helali. The *Shriveling world* application is available here: https://bit.ly/shrivelApp (accessed on 13 September 2018). Open source code can be accessed here: https://github.com/theworldisnotflat/shriveling_world (accessed on 13 September 2018). The map of this article was created by using the version v0.9-alpha of *Shriveling world*: http://bit.ly/shrivelZen (accessed on 13 September 2018), with post-treatments in Blender (https://www.blender.org/) (accessed on 13 September 2018).

The Chinese geographical time-space in 2006 is characterised by two transport modes. Road transport is modelled with a uniform speed of 100 km per hour which is consistent with the measurement of travel times by road between major Chinese cities. The air transport system is divided in two sets of links. According to our analysis (see Section 3), long range flights, beyond 2000 km, follow a speed of 750 km per hour. Lower range flights have a decreasing speed following their length in kilometres. The Chinese geographical time-space is generated by the confrontation of long range flight speed (750 km/h) and road speed (100 km/h). The slope of cones, where cars circulate, is the same for all the geographic surface. The road system being modelled as an anisotropic surface with uniform speed, physical geography constraints like mountains and rivers are completely ignored in the representation. The geometry of cones and is computed (see Appendix A) from the fundamental ratio of the maximum speed divided by road speed giving the value of 7.5.

The implementation of the geographical time-space model to China has obeyed a series of choices and parameters that we will expose and justify now.

- On this representation, no geographical projection is used: Cities are located on the terrestrial sphere which explains that the fastest arcs, in red, are not straight lines but curves as they follow the geodesic.
- An angle of 45 degrees between the camera axis and the tangent to the surface of the earth, has been chosen as a compromise between the readability of the final image, and the preservation of proportionality of north-south compared to east-west lengths.
- Lights have been introduced in order to maximise the readability of the three-dimensional structure: A directional light is completed by punctual lights between the cones in order to improve contrasts.
- Cones are set in white colour with shadows to express their three-dimensional nature.
- Aerial edges are coloured in two colours according to the categories of long and short haul. Long-haul speed, i.e., the fastest observed speed on this geographical space, determines the parameters of the geometry of all representations: Cones slope, and the geometry of the other edges; given their importance we attributed them a highly visible colour, red. Slower edges, on short haul routes, are coloured in green to highlight their relatively weaker performance in geographical time-space.

This map is the first representation of the Chinese geographical time-space in relief, and is also the first map ever with this new three-dimensional model with cones and rounded edges.

On the representation, the speed of the road transport is expressed through the slope of cones; the idea that the road allows access to all space is expressed by the fact that the cone is a surface. The long range links of the air transport system are the fastest transport mode available: They form direct geodesic lines between cities, shown in red. Slower, shorter range flights have length proportional to the time needed to connect cities, and expand above the geodesic line and are represented in green. If the air transport system is dramatically efficient for reducing time-space on long ranges, its efficiency drops on shorter distances. To such extent that fast terrestrial modes can compete with air transport

on many city pairs, as shows the development of the high-speed rail in China since the end of the first decade of the 21st century.

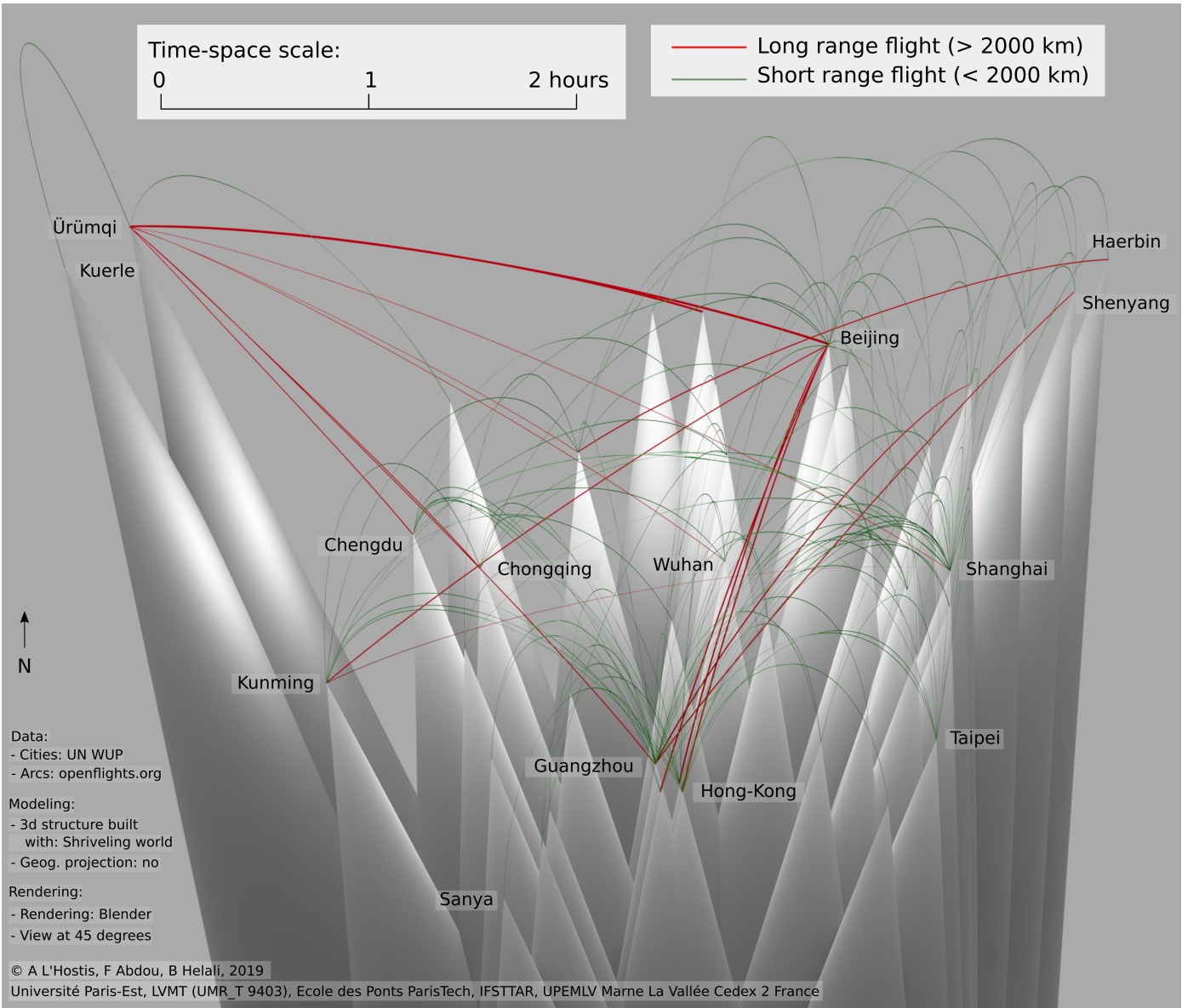

**Figure 4.** A representation of the Chinese geographical time-space in 2006. View of a model generated by the *Shriveling world* software, unprojected Chinese cities, flight information from openflights.org, UN WUP cities data.

The representation opposes the efficiency of the air transport system in crossing time-space and the relative slowness of terrestrial modes. Considering that most movements at the scale of the country will involve combining flying and driving, the map proposes a coherent representation of time-space for such trips, where a time-scale for approximate measurement is available. The coherence between the measurements of time-space and the proposed graphic representation is not observed on the usual map or on other types of anamorphic cartography [49].

In exchange for the theoretical benefits it provides, from a general point of view, this implementation has engaged the need to address a whole new range of issues, some of them unprecedented in cartography. The main issues identified so far are listed here:

- The need to consider the phenomenon of occultation of three dimension shapes.

- The conflict between the desire to rotate the structure in all directions—a common practice in three-dimensional modelling–and the convention of cartography that invite to restrict it to rotate exclusively on an east-west axis.
- The combination of geographical projection, from the geoid to a two-dimensional flat surface, with other types of projection used to transform a three-dimensional scene into a computer screen image.
- The need for shadings to reveal three-dimensional shapes that may conflict with the greying conventions of cartography used to express terrestrial relief, and conventional colour choices in cartography [50].
- The need for a background, which is not generally dealt with in cartography.
- The need to adjust the graphical parameters of networks–width, colour, shape, and transparency–to allow for reading the three-dimensional surface situated below.
- The treatment of terrestrial and maritime borders.

The model exposed in this paper is a proposition that aims at solving these issues. However many alternatives can be imagined, and will be investigated in further work. This could include projecting different GIS layers onto the three-dimensional surface in order to facilitate the reading of the representation. In addition, the current implementation covers a single large country; new issues may occur when considering other smaller or larger geographical entities.

## 9. Conclusions

We introduced a new model of geographical time-space representation aimed at improving and developing a model proposed in the 1990s. Our aim has been to rationalise the production of the maps, and to improve readability. Compared to previous representations of the same type, the novelty of the model resides in the use of cones to describe the terrestrial surface instead of graph faces, and in the use of curves instead of broken segments for edges.

The higher complexity of the three-dimensional model has necessitated to writing a new code for generating the three-dimensional structure, and to using existing three-dimensional rendering software.

The approach lies at the intersection of two domains involving graphic representation i.e. cartography and three-dimensional computer graphics, that, with the exception made of the implementation of time-geography in GIS [13,51,52], have rarely been fully combined: We intend to respect as far as possible the rules of cartography, but we also want to take advantage of the potential of three-dimensional computer graphics to improve the readability of the final image. Consequently, the representation produced should be understood as borrowing to the two domains.

We implement the model on the Chinese space, for the first time. The Chinese geographical time-space in the year 2006 is produced by the combination and the confrontation of the fast air transport system and of the 7.5-times slower road transport system. The very steep slope of cones expresses the relative difficulty of travelling in the terrestrial time-space, as well as the comparably extreme efficiency of long-range flights for movement between cities. The image also shows that the very high efficiency of the air transport for crossing geographical time-space is not met on most origin-destination pairs in the Chinese space: Many flights follow a long curve above the earth surface that account for their relative inefficiency, i.e., the relatively slow speed of short haul flights, as is the case between Beijing and Shanghai (Beijing-PEK and Shanghai-SHA airports are 1075 km by great circle distance away and connected in 2h15 resulting in a commercial speed of 478 kph). This situation highlights the relevance of the current development of high-speed rail in China. Finally, the whole image proposes a coherent representation of the geographical time-space where fast city-to-city transport is combined with slow terrestrial systems that allow one to reach any location.



The major challenge posed by geographical time-space three-dimensional representation lies in the issue of readability. Proposing a coherent geographical time-space representation that moves away from conventional cartography and that is fully based on the three dimensions entails the introduction of an unprecedented set of cartographic and graphical issues. The developments exposed here–the cones and the curved edges–are two proposals in this direction of a more intelligible representation. Much remains to be done, in a domain were graphical solutions should meet cartographic constraints. Once a suitable level of legibility is obtained, the next steps should involve users testing in order to assess the improvement of the representation over previous designs, and validate it beyond the theoretical arguments provided here.

**Author Contributions:** Conceptualisation, Alain L'Hostis; methodology, Alain L'Hostis and Farouk Abdou; software, Farouk Abdou; data curation, Alain L'Hostis and Farouk Abdou; writing—original draft preparation, Alain L'Hostis; writing—review and editing, Alain L'Hostis and Farouk Abdou; visualisation, Alain L'Hostis and Farouk Abdou. All authors have read and agreed to the published version of the manuscript.

**Funding:** The APC was funded by the University Gustave Eiffel.

**Data Availability Statement:** Two sources of data are needed to produce the representation. We used the 2014 revision of the World Urbanisation Prospect by the United Nations [48] for the cities. The air services network is derived from the data provided by openflights.org representative of the year 2006 (https://openflights.org/data.html).

**Acknowledgments:** The authors thank the late Waldo Tobler for his encouragements and insightful remarks on earlier versions of the map. The authors thank Jean-François Hangouët and Antoine Pinte from IGN, Nicolas Roelandt, Marielle Cuvelier from Université Gustave Eiffel, and Jules L'Hostis for helping to solve the trigonometric equations. The authors thank Billel Helali for his work on the code generating the three-dimensional structure, and for his work on the parameters of the rendering.

**Conflicts of Interest:** The authors declare no conflict of interest.

## Appendix A. Cones Parameters in Projected Geometry

Considering the geometry of cones as shown in Figure A1, we have the maximum speed as the *ab* straight line and the road speed along the cone in *am*. Cities *a* and *b* are linked by a fast transport mode and by a terrestrial transport mode characterised by a lower speed. By construction of the time-space map, the length of links *ab* by fast transport and *amb* by slow transport are proportional to the duration of the respective trips.

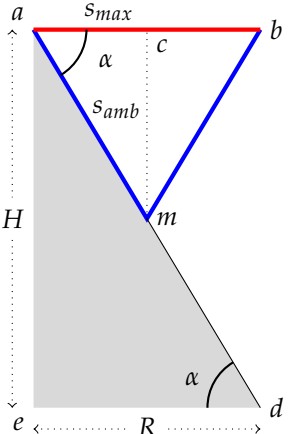

**Figure A1.** Cones and edges in projected geometry.

The representation indicates the duration $t$ of trips by measuring the length of segments. Hence, a fixed factor $k$ corresponding to a time-space scale links duration and length so that $t_{ab} = k \cdot l_{ab}$ and $t_{am} = k \cdot l_{am}$. Time, length, and speed are linked by the simple formula of $t_{ab} = l_{ab}/s_{max}$, but on the segment $am$ the duration of the trip corresponds to the length $ac$ travelled at the speed on $ab$ and hence $t_{am} = l_{ac}/s_{amb}$. From this we obtain a formula linking the length $ac$ and $am$ to a ratio of speed:

$t_{ac} = k \cdot l_{ac}$

$l_{ac}/s_{max} = k \cdot l_{ac}$

$k = 1/s_{max}$

$t_{am} = l_{ac}/s_{amb} = k \cdot l_{am}$

$l_{ac}/s_{amb} = l_{am}/s_{max}$

$l_{am} = l_{ac}\frac{s_{max}}{s_{amb}}$

The slope is then given by the ratio between speed $s_{max}$ on $ac$ and the speed $s_{amb}$ on $am$.

Applying the trigonometric formula $1 + \tan(\alpha)^2 = \frac{1}{\cos(\alpha)^2}$ with $\tan(\alpha) = \frac{H}{R}$ and $\cos(\alpha) = \frac{s_{max}}{s_{amb}}$ gives $1 + \frac{H^2}{R^2} = \frac{1}{(\frac{s_{max}}{s_{amb}})^2}$ and finally:

$$R = \frac{H}{\sqrt{(\frac{s_{max}}{s_{amb}})^2 - 1}}.$$

This final formula links the radius of the cone to its height and to the ratio between maximum speed and speed along the slopes of the cone. For a given radius, the height of the cone will follow Formula (A1), according to the speed differential:

$$H = R\sqrt{(\frac{s_{max}}{s_{amb}})^2 - 1}. \tag{A1}$$

**Appendix B. Cones and Straight Edges in Spherical Geometry**

In spherical geometry, the fastest edge is not a straight euclidean line, but rather a geodesic, or great circle edge between two cities, following the curvature of the earth. In the simplest variant of the model, a slower edge is drawn as two chained rectilinear segments of equal length, in the plane formed by the two cities orthogonal to the surface of the earth. The general principle of the model states that the length of the edge $amb$, in Figure A2, is proportional to the length of the fastest speed edge $g$, hence to the ratio of speed.

The length of the arc $g$ between $a$ and $b$ is given by the formula $g = r\theta$, and hence: $am + mb = \frac{s_{max}}{s_{amb}}g = \frac{s_{max}}{s_{amb}}r\theta$ $am + mb = \frac{s_{max}r\theta}{s_{amb}}$ $am = \frac{s_{max}r\theta}{2s_{amb}}$.

We need to determine the coordinates of point $m$ and the length of segment $om$:

$$om = r - (cd + dm).$$

In case the edges are drawn above the surface of the earth, then the length of segment $om'$ is:

$$om' = om + 2dm = r - cd + dm.$$

Considering the circle centred in $a$ and having $am'$ for radius we have:

$$x^2 + y^2 = r^2.$$

The point $m$ has for coordinate $x = ad = ab/2$ and $y = dm$ then:

$$(ab/2)^2 + dm^2 = am^2$$

$$dm = \sqrt{am^2 - (ab/2)^2}$$

$$dm = \sqrt{\left(\frac{s_{max}r\theta}{2s_{amb}}\right)^2 - (ab/2)^2}.$$

Considering the triangle $adO$:

$$\sin\frac{\theta}{2} = ad/r = ab/2r$$

$$ab = 2r\sin\frac{\theta}{2}.$$

We then update the previous equation as:

$$dm = \sqrt{\left(\frac{s_{max}r\theta}{2s_{amb}}\right)^2 - (r\sin\frac{\theta}{2})^2}$$

$$dm = r\sqrt{\left(\frac{s_{max}\theta}{2s_{amb}}\right)^2 - \sin^2\frac{\theta}{2}}.$$

Considering the two points $m$ and $m'$ we will now focus on the general case of $m'$, knowing that $dm = -dm'$:

$$om = r - (cd + dm)$$

$$om' = r - cd + dm.$$

Considering the trigonometric formula of the height of the chord between $a$ and $b$:

$$cd = r - \sqrt{r^2 - \frac{ab^2}{4}}$$

$$om' = r - \left(r - \sqrt{r^2 - \frac{ab^2}{4}}\right) + dm$$

$$=> om' = \sqrt{r^2 - \frac{ab^2}{4}} + dm.$$

Considering the trigonometric formula of the height of the chord between $a$ and $b$:

$$cd = r - \sqrt{r^2 - \frac{ab^2}{4}}$$

$$om' = r - \left(r - \sqrt{r^2 - \frac{ab^2}{4}}\right) + dm$$

$$=> om' = \sqrt{r^2 - \frac{ab^2}{4}} + dm.$$

Considering the formula of the length of the chord $ab = 2r\sin\frac{\theta}{2}$ we have:

$$om' = \sqrt{r^2 - r^2\sin^2\frac{\theta}{2}} + dm = r\sqrt{1 - \sin^2\frac{\theta}{2}} + dm$$

$$om' = r\cos\frac{\theta}{2} + dm$$

$$om' = r\cos\frac{\theta}{2} + r\sqrt{\left(\frac{s_{max}\theta}{2s_{amb}}\right)^2 - \sin^2\frac{\theta}{2}}.$$

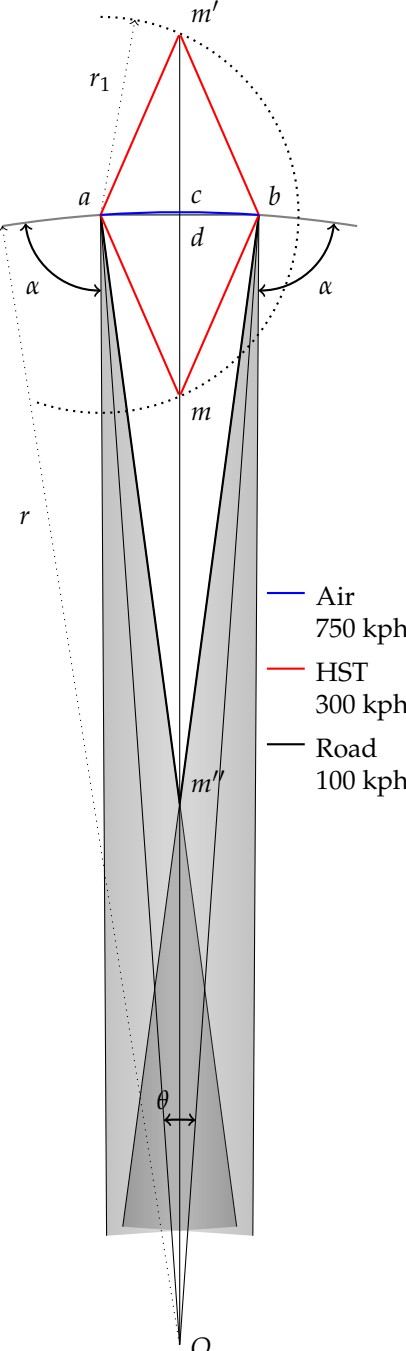

**Figure A2.** Drawing edges and cones with different speed in the spherical geometry.

Here comes the Formula (A2) of the length of the segment $om'$, in the general case, as a function of speed, $r$, and $\theta$ as:

$$om' = r\left(\cos\frac{\theta}{2} + \sqrt{\left(\frac{s_{max}\theta}{2s_{amb}}\right)^2 - \sin^2\frac{\theta}{2}}\right). \tag{A2}$$

In order to account for the lower commercial speed of aircraft services on short distances, as introduced before [28], we consider a cruise speed of 750 km per hour beyond 2000 km of range, and the following formula below 2000 km:

$s_{amb} = (s_{long-range}/2000)g = 0.375g$ with $g$ as the length of the geodesic minimum path. Considering that $g = r\theta$, we have:

$$s_{amb} = 0.375r\theta$$

$$om' = r\left(\cos\frac{\theta}{2} + \sqrt{\left(\frac{s_{max}\theta}{0.75r\theta}\right)^2 - \sin^2\frac{\theta}{2}}\right).$$

In the case of aircraft links with length below 2000 km the formula of the length of the segment $om'$ becomes the following :

$$om' = r\left(\cos\frac{\theta}{2} + \sqrt{\left(\frac{s_{max}}{0.75r}\right)^2 - \sin^2\frac{\theta}{2}}\right). \tag{A3}$$

Finally we provide the formula of the midpoint between two given geographical locations with $\phi$ as latitude, $\lambda$ as longitude, and $\Delta\lambda = (\lambda_2 - \lambda_1)$:

$$Bx = \cos\phi_2\cos\Delta\lambda$$

$$By = \cos\phi_2\sin\Delta\lambda$$

$$\phi_m = \arctan 2$$

$$\left(\sin\phi_1 + \sin\phi_2, \sqrt{(\cos\phi_1 + B_x)^2 + B_y^2}\right) \tag{A4}$$

$$\lambda_m = \lambda_1 + \arctan 2(B_y, \cos\phi_1 + B_x).$$

**Appendix C. Drawing Smoothed Edges in Projected Geometry**

In the three-dimensional time-space, the length of edges is proportional to the time needed to travel between cities.

In Figure A3 we provide several graphical solutions to the constraint of drawing an edge with given length between cities *A* and *B*. The straight segments solution geometry is described in Appendix B and is shown here in a dotted black line. Four Bézier curve solutions are represented with their respective control points. The Bézier curve with one single control point is shown in red. The blue and yellow Bézier curves with two control points differ in their extremities. The yellow curve follows a vector orthogonal to the earth surface in *A* and in *B*. Finally, a Bézier curve with four control points allows a solution that tangents the earth surface in start and end points.

The proposed solutions are graphical variations respecting the general principle of the representation. They allow to explore the graphical possibilities in terms of edges layout. Smoothed curves avoid to graphically highlight the midpoint of the straight segments, as in [28], which has no significant geographical meaning. Orthogonal or tangent starting and ending curves propose different ways to graphically highlight cities or transport nodes which bears true geographical meaning.

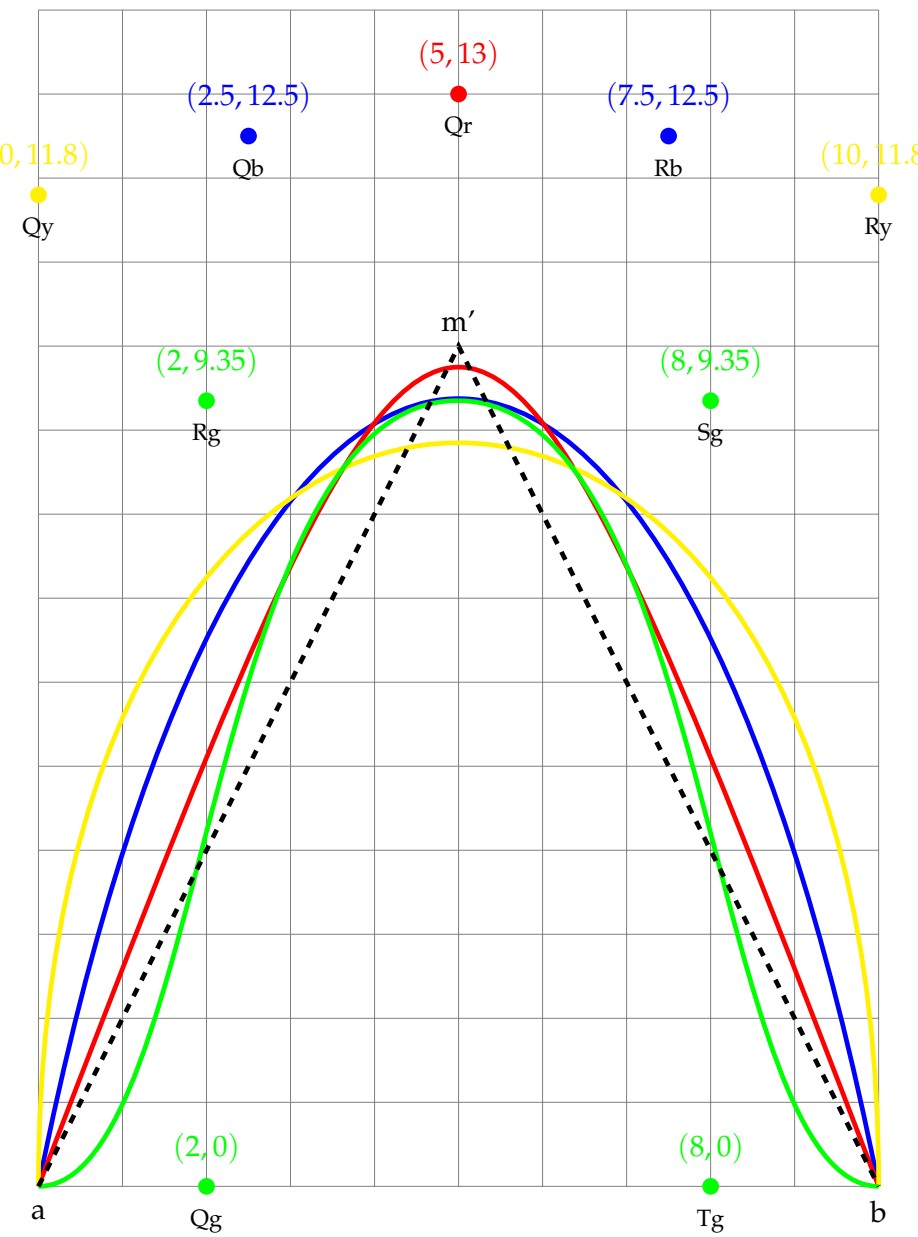

**Figure A3.** Several graphical solutions to the constraint of drawing an edge with a given length between cities *a* and *b*: Two straight segments (dotted) and different Bézier curves (red with one control point *Qr*; blue with two control points *Qb* and *Rb*; yellow with two control points *Qy* and *Ry*; and green with four control points *Qg*, *Rg*, *Sg*, and *Tg*). The position of control points was adjusted to constrain the curve length.

### Geolocation Information

The representation of the geographical time-space in relief covers:

- People's Republic of China.
- Taiwan.

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

## Short Biography of Authors

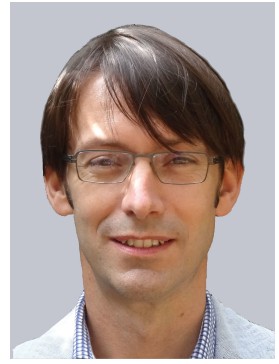

**Alain L'Hostis** Alain L'Hostis is senior researcher at the City, Mobility, Transport laboratory (LVMT) from the Université Gustave Eiffel. He holds a PhD and an habilitation thesis in spatial planning. His research topic is about geographical distances: he works on time-space cartography, on inter-urban distances through the measurement of accessibility, and on intra-urban distances in relation with the urban model of TOD.

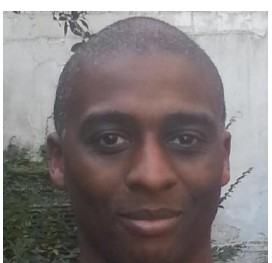

**Farouk Abdou** Farouk Abdou works in Ministère de la Défense as project leader in software development.