# Peer review of "What Is the Shape of Geographical Time-Space? A Three-Dimensional Model Made of Curves and Cones"

_ijgi, doi:10.3390/ijgi10050340_

Round 1

Reviewer 1 Report

Overall, the manuscript addresses a relevant and original topic, although some supplementations and improvements are required. That is why, I would like to list of major and minor remarks below.

General remarks (major):

- The abstract is very confusing and not clear. Kindly revisit it carefully in light of the comments given.  It must include the following: i) study background and motivation, ii) aims and objective of the current study, iii) data used and methods adopted, and iv) specific conclusions are drawn from the analysis;

- There are several grammar mistakes and typos that need to be corrected.  Also there are many mistakes in punctuation. In my opinion, a proofread of the manuscript is necessary;

- The author(s) use data from 2006 and 2014 (lines 232 -244), while the article was submitted in 2021. So the analysis considers situation of chosen aerial services noticed 15 years ago. The justification for the use of older data is then essential;

- The discussion may be also differentiated in a separate section.

Details (minor):

- lines 102 – 104 – a source or reference needed;

- line 120 – inappropriate type of reference (and 90 appears far too early);

- Figure 3 – the author(s) used symbols and formulas, which are not explained (we can only guess that ‘a’ and ‘b’ represent cities) – supplementation highly recommended;

Author Response

IJGI reply 1 April 20211 Reviewer 1Overall, the manuscript addresses a relevant and original topic, although somesupplementations and improvements are required. That is why, I would like tolist of major and minor remarks below.1.1 General remarks (major):The abstract is very confusing and not clear. Kindly revisit it carefullyin light of the comments given. It must include the following: i) studybackground and motivation, ii) aims and objective of the current study,iii) data used and methods adopted, and iv) specific conclusions are drawnfrom the analysis; The abstract is re-written based on the proposed indications: ”Ge-ographical time-spacesexhibit a list of properties, including spaceinversion, that turns any representation effort a complex task. Inorder to improve the legibility of the representation and leveragingthe advances of three dimensional computer graphics, the aim of thestudy is to propose a new method extending time-space relief cartog-raphy introduced by Mathis and L’Hostis . The novelty of the modelresides in the use of cones to describing the terrestrial surface insteadof graph faces, and in the use of curves instead of broken segments foredges. We implement the model on the Chinese space. The Chinesegeographical time-space of the reference year 2006 is produced by thecombination and the confrontation of the fast air transport systemand of the 7.5 times slower road transport system. Slower, shortrange flights are represented as curved lines above the earth surfacewith longer length than the geodesic, in order to account for a slowerspeed. The very steep slope of cones expresses the relative difficultyof crossing terrestrial time-space, as well as the comparably extremeefficiency of long-range flights for moving between cities. Finally, thewhole image proposes a coherent representation of thegeographicaltime-spacewhere fast city to city transport is combined with slowterrestrial systems that allow to reach any location.”1

There are several grammar mistakes and typos that need to be corrected.Also there are many mistakes in punctuation. In my opinion, a proofreadof the manuscript is necessary;A search for typos, grammar mistakes and punctuation errors hasbeen conducted throughout the text.The author(s) use data from 2006 and 2014 (lines 232 -244), while thearticle was submitted in 2021. So the analysis considers situation of chosenaerial services noticed 15 years ago. The justification for the use of olderdata is then essential;Several elements of discussion about this point are added in section 7on the dataset: Regarding the list of cities, 2014 dataset is ” the mostrecent urban high quality dataset available at the time of creationof the representation”. Concerning the flight data, ”the network offlights is from the year 2006, the most recent data-set available fromthis source. In addition, the aerial transport system of 2006 can beconsidered as representative of the 2014 condition.”The discussion may be also differentiated in a separate section.The discussion about what the model says is combined in this sec-tion with methodological issues. The general purpose of the article isto expose a new type of representation, and hence comments on themodel combine the two aspects, geographical insights and method-ological developments.1.2 Details (minor):lines 102 – 104 – a source or reference needed;a reference has been added here (chenDevelopmentHistoryAccessibil-ity2018)line 120 – inappropriate type of reference (and 90 appears far too early);90 refers to a page number of ref number 37, formatting according tothe revue templateFigure 3 – the author(s) used symbols and formulas, which are not ex-plained (we can only guess that ‘a’ and ‘b’ represent cities) – supplemen-tation highly recommended;explanation of symbols are introduced in the adjacent text

Reviewer 2 Report

  1. The paper should be improved by answering this question: What presented model for transport network speed visualization in 3D cones gives more for transport network planning that transport network speeds for flights and road transport presented table.
  2. It would be very valuable approach to present this 3D model time-space curves and cones in GIS environmental to use base maps, real coordinates, probably terrain dataset.
  3. List of cited literature could be newer, just one citation is from 2019 year.
  4. I think necessary to say some words about barriers (mountains, water resources and etc) in road network routes. Are they evaluated in model?

Author Response

1
Reviewer 2
1. The paper should be improved by answering this question: What pre-
sented model for transport network speed visualization in 3D cones gives
more for transport network planning that transport network speeds for
flights and road transport presented table.
• The aim of the representation is to express the shape of past, cur-
rent and future geographical time-space; its purpose is not directly
to provide analyses, reflections and projections regarding transport
planning issues. It may be – and has been – used to inform spatial
and transport planning processes, but its main aim is more heuristic
2. It would be very valuable approach to present this 3D model time-space
curves and cones in GIS environmental to use base maps, real coordinates,
probably terrain dataset.
• The idea of ”projecting different GIS layers onto the three-dimensional
surface in order to facilitate the reading of the representation” is in-
troduced as a future possible development just before the conclusion,
page 8.
3. List of cited literature could be newer, just one citation is from 2019 year.
• Where already present in the bibliography:
– 2016 1
– 2017 4
– 2019 1
In order to address the call for a newer reference list, relevant new
sources have been added:
– 2018 2
– 2020 1
14. I think necessary to say some words about barriers (mountains, water
resources and etc) in road network routes. Are they evaluated in model?
• A footnote in section 8 is added to cover this point : ”The road system
being modelled as an anisotropic surface with uniform speed, phys-
ical geography constraints like mountains and rivers are completely
ignored in the representation.”

Reviewer 3 Report

This paper proposes a geographical time-space model for the three dimensional representation of geographical time-space attributes. The model was developed in terms of the extending time-space relief cartography, and cones and curves were used to identify the terrestrial surface. The model was applied in the representation of Chinese air and road transport system. The varied speed of transports with different modes and the sophisticated crossing terrestrial time-space can be explained using the developed model. 

Major comments

(1) Introduction. Can you please add a few paragraphs to review and summarise previous methods of time-space representation? The theoretical and practical basis should be clearly demonstrated before developing a new method.

(2) Figure2: Can you please consider using an exponential function to fit this figure?

Round 2

Reviewer 1 Report

 Thank you for your careful revisions. Most of my comments from first round review were addressed. The paper has a significant improvement.

I have found a few minor editing and punctuation errors – it would be necessary to check the text one more time to correct them.

Reviewer 2 Report

In my opinion the paper could be accepted.

Reviewer 3 Report

My concerns have been addressed in the current version.